# Addressing the Knowledge Deficit in Hospital Bed Planning and Defining an Optimum Region for the Number of Different Types of Hospital Beds in an Effective Health Care System

**DOI:** 10.3390/ijerph20247171

**Published:** 2023-12-12

**Authors:** Rodney P. Jones

**Affiliations:** Healthcare Analysis & Forecasting, Wantage OX12 0NE, UK; hcaf_rod@yahoo.co.uk

**Keywords:** hospital bed numbers, optimum occupancy, bed models, deaths, international comparison, benchmarking, healthcare policy, policy-based evidence, patient flow, queuing theory, infections, surge capacity

## Abstract

Based upon 30-years of research by the author, a new approach to hospital bed planning and international benchmarking is proposed. The number of hospital beds per 1000 people is commonly used to compare international bed numbers. This method is flawed because it does not consider population age structure or the effect of nearness-to-death on hospital utilization. Deaths are also serving as a proxy for wider bed demand arising from undetected outbreaks of 3000 species of human pathogens. To remedy this problem, a new approach to bed modeling has been developed that plots beds per 1000 deaths against deaths per 1000 population. Lines of equivalence can be drawn on the plot to delineate countries with a higher or lower bed supply. This method is extended to attempt to define the optimum region for bed supply in an effective health care system. England is used as an example of a health system descending into operational chaos due to too few beds and manpower. The former Soviet bloc countries represent a health system overly dependent on hospital beds. Several countries also show evidence of overutilization of hospital beds. The new method is used to define a potential range for bed supply and manpower where the most effective health systems currently reside. The method is applied to total curative beds, medical beds, psychiatric beds, critical care, geriatric care, etc., and can also be used to compare different types of healthcare staff, i.e., nurses, physicians, and surgeons. Issues surrounding the optimum hospital size and the optimum average occupancy will also be discussed. The role of poor policy in the English NHS is used to show how the NHS has been led into a bed crisis. The method is also extended beyond international benchmarking to illustrate how it can be applied at a local or regional level in the process of long-term bed planning. Issues regarding the volatility in hospital admissions are also addressed to explain the need for surge capacity and why an adequate average bed occupancy margin is required for an optimally functioning hospital.

## 1. Background

By way of background, the National Health Service (NHS) in the UK is run in a devolved manner, such that each of the four countries in the Union determines the level of funding, sets their own policies, and manages the NHS in different ways. This author entered the English NHS in the early 1990s and was quickly involved in planning for the construction of a new hospital. I was informed that the ‘correct’ way to forecast future bed numbers was to take age-banded current admission rates (admissions per head of population) and use forecast population demographic changes (age-based forecasting) to estimate future inpatient admissions and to multiply such future admissions by the anticipated future average length of stay (LOS) to arrive at occupied bed days, and then apply an occupancy margin to arrive at the number of available beds [1]. My experience with bed planning then led to a 30-year search for more reliable methods, with many related published studies.

Throughout this study, a body of nearly 300 supporting publications will not be referenced, as it is assumed that the reader will make themselves aware of the scope of this wider evidence. To facilitate access to this material, the full list is given in Appendix A. This material is broken down into 19 alphabetically labeled sections (A to S), with papers in each section having a number. The reason that there are 19 sections is that almost all the issues are interlinked. Hence, capacity risk is linked to financial (cost) risk. Volatility in admissions is central to capacity and cost risk, but also to understanding the optimum bed occupancy rate, etc. In the text, the reader will be referred to the relevant section heading, i.e., A to R, or to a specific reference such as Appendix A, etc.

It should be noted that the British Journal of Healthcare Management (BJHCM) acknowledged the need for an extended research series in this poorly explored area. As part of this research, BJHCM ran a monthly series called “Money Matters”, consisting of short mini-research pieces that explored the far wider issues of why it was so hard for healthcare organizations to achieve both bed capacity and financial balance.

Additional references will be made to the wider body of literature using the usual bracketed reference number system. One example is the excellent systematic review of the current state of hospital bed modeling by Ravaghi et al., published in 2020 [2], which identified 11 models and 5 methods. It should be noted that some of these models/methods involve short-term or tactical forecasting, while this study focuses on long-term bed planning.

I very quickly realized that this method was entirely unreliable and seemed to be deemed ‘correct’ simply because it grossly underestimated future demand, thereby ‘confirming’ the political view that England needed fewer hospital beds because the NHS was ‘inefficient’. The abject failure of age-based forecasting to predict future admissions is illustrated in Appendix A [3,4,5,6,7,8,9,10], where the actual trend in admissions for diseases of the appendix (mostly appendicitis) is compared to the age-based forecast.

It has subsequently taken around 30 years to fully understand the intricacies of long-term bed forecasting. Any aspect of a study taking 30 years is likely to indicate that the underlying issues involve a profoundly complex system. The huge divergence between actual and age-based forecasts for diseases in the appendix to Appendix A is but one example of such complexity.

An additional aim is to detail how international benchmarks and readily available data can be applied to the local level of hospital bed planning. The important role of volatility in admissions and bed demand will be discussed in terms of the issue of surge capacity and the average bed occupancy. The profoundly important role of government policy will be illustrated. Lastly, it is hoped that this study will serve as a technical resource for further research, hence the recourse to an extended series of Appendix A.

Each section will discuss the limitations of current methods and suggest suitable alternatives. Many of the methods/data sources have already been covered; however, where an alternative is suggested, the methods/data sources will be given in that section.

## 2. Introduction

It must be pointed out that the then-accepted method of long-term bed forecasting using population age structure [1] has never been proven to work per se, see Appendix A. While it is true that age-standardization is a valid tool in the field of epidemiology [11], it is mainly employed in a retrospective way and requires types of time series regression or other forecasting to extrapolate into the future. Indeed, age-banded population forecasts are themselves uncertain and contain numerous assumptions [12,13,14]. My own experience is that both births and deaths, which are fundamental to population change, show profoundly high uncertainty, see Appendix A, indicating ambiguity in the assumptions or that the hidden assumptions show intrinsically high variability.

The huge deficiency in this method is that the admission rate per head of population is assumed to stay constant, while the reality is that it changes (mainly increases) over time [1], also see Appendix A, as the scope of medical interventions grows, hence the threshold to admission declines due to the realization that new interventions can extend lifespan. The proportion of elderly people living alone, and the availability of caregivers have a big impact on the ensuing hospital admissions [15,16]. The rate of increase also depends on the admission type; namely elective versus emergency [17]; and the specialty and specific intervention—as for diseases of the appendix; etc. In theory, this eventuality was supposed to be covered by an assessment of the likely impact of new technology [18], but in practice, this was always underestimated.

Indeed, the reality was that although the English NHS was operating with fewer and fewer available beds, the average overnight stay bed occupancy rate was rapidly rising (with associated operational problems) [19,20], also Appendix A, and the number of occupied beds had not changed since 1998/1999, see Appendix A—although the models said they were supposed to be declining.

My search eventually led to the interesting observation that occupied bed numbers appeared to be strongly influenced by the absolute number of deaths, see Appendix A, which is termed the nearness-to-death (NTD) or time-to-death (TTD) phenomena. NTD/TTD is a very well-recognized concept in the field of economics and the forecasting of future healthcare costs [21,22]. While NTD/TTD has been extensively researched for over 30 years, it has never been incorporated into bed modeling.

This seemingly contradictory finding to the currently accepted views regarding ‘correct’ bed modeling arises for several reasons:Around 55% of a person’s lifetime hospital bed usage occurs in the last year of life, irrespective of the age at death [23,24,25].Increased lifespan expands the opportunity for lifetime hospital admissions, and hence the 55% figure (#1 above) leads to a steady inflation of bed demand with increasing longevity [25,26].About half of medical bed occupancy appears to be influenced by fluctuations in the absolute number of deaths, seemingly due to local and national outbreaks of infectious agents. On this occasion, deaths serve as a proxy for the specific and nonspecific effects of wider infections, mediated by inflammatory and other processes, necessitating hospital admission but not leading to immediate death (see Section 11). The COVID-19 pandemic is an excellent example of this relationship, where there are far more admissions than deaths.In an aging population with increasing multimorbidity, the ‘at-risk’ population [27], where fluctuations in the absolute number of deaths are serving as a proxy for morbidity, is also increasing with time.In deprived/poorer areas, the population tends to die at a younger age [26], hence bed usage is moved forward in time due to #1 above.As a first approximation, because everyone must die, the simple division of currently occupied bed days by the current number of deaths seems to work remarkably well, see Appendix A. This approximation will break down if the ratio of births to deaths substantially changes and relies on a balance between rising admissions per death (contingent on the issues raised above regarding the elderly living alone and the availability of caregivers) and declining average length of stay (LOS). In England, admissions seem to be rising roughly as fast as the length of stay is declining.

Having established why NTD/TTD is central to bed demand, the utility of the model will now be discussed before progressing to how the new model can be extended to the regional and local levels within a country. The role of infectious outbreaks on the intrinsically high volatility in admissions (especially emergency) will be covered in Section 11, where issues of surge capacity and the average bed occupancy margin needed for optimum performance will be explored—especially in the context of local health care capacity.

## 3. Additional Supporting Analysis

The methods used to analyze the international data and the incremental development of the model have been described previously, see Appendix A. Additional supporting analysis is also given in this review, both in the text and in the Appendix A. All data are publicly available [28,29,30,31,32,33,34,35,36,37,38], and the source is cited alongside the respective Figure or Table. A description of any methods is included in each section. Where necessary, a Appendix A has been added.

## 4. The New Model

### 4.1. Background to the Model

For any model of international bed comparison to be realistic, it needs to be based on readily available international data. I first became aware of the NTD/TTD phenomena around 2010 and suggested that this was the missing ingredient in bed modeling, see Appendix A. Then, followed two studies to demonstrate the feasibility of the concept, see Appendix A. The first international comparison was published in 2018, see Appendix A. Thereafter followed the incremental development of the model, mainly around the slope and then the relationship between the slope and the intercept.

In the absence of prior knowledge of the intricacies of each country, the outputs from the model depended on the allocation of countries into broad groups (poor through very high). This process was conducted using initial visual observation and then repeated regression studies after moving marginal countries between groups. The maximization of R-squared was used to achieve the final allocation. This process was also conducted in the absence of the potential role of the age-standardized mortality rate (ASMR), detailed in Section 8, on the outputs. It is suggested that future development of the model involve ASMR as a variable or be used to construct a wider range of groups.

In the new model, the ratio of beds per 1000 deaths is plotted against the ratio of deaths per 1000 population. The latter is called the crude mortality rate (Figure 1) and is an approximate (and widely available) measure of the population age structure. When international bed data are plotted using this method, an infinite series of lines of equivalent bed availability can be applied as a logarithmic relationship. The lines of equivalent bed availability reflect the different crude mortality rates between countries.

The method has been applied to investigate disparate bed numbers in the states of Australia, the USA, and India, see Appendix A, for both total curative, see Appendix A, medical, see Appendix A, and critical care beds, see Appendix A, and to investigate the expressed need for beds revealed as ‘occupied’ beds in Australian states and English Clinical Commissioning Groups (CCGs), see Appendix A. In the case of Australian states, adjustment was applied for the proportion of indigenous people (with known higher levels of poor health), in Appendix A, and for the differing levels of social deprivation experienced between English CCGs, in Appendix A. It was finally discovered that the intercept and slope of the logarithmic relationships are directly linked, such that the lines of equivalent bed availability can be defined using the value of the intercept as shown in Figure 1, see Appendix A.

Countries lying along or near a line of equivalence have similar bed numbers; however, the reasons for equivalence can be many and varied. It has been noted that the average bed occupancy for a country depends greatly on the average size of the constituent hospitals, in Appendix A, and that acute bed demand may depend on the wider availability of nursing home beds, in Appendix A, and factors such as the implementation of integrated care [39,40,41]. A further extension of the model was to define a likely feasible region for the optimum number of critical care beds in the absence of excessive ‘futile’ intervention, which can occur in fee-for-service health systems, see Appendix A.

Figure 1 contains data for world countries in 2019 (just before the disruption caused by the COVID-19 pandemic) along with several lines of equivalent available beds. In Figure 1, the number of beds in each country is in 2019 or the nearest year after extrapolation of the recent trend to 2019, by either linear or polynomial regression.

In terms of total beds, most countries are trending down or stationary, but a few, such as China and Türkiye, are trending upward. World countries have been grouped from the world’s poorest through those with very high available beds—once again for many and varied reasons. Japan and South Korea are excessively high because they seemingly (incorrectly) count ‘nursing home’ type care as a ‘curative’ bed, see Appendix A. The data for North Korea are also very high but has probably been fabricated. Germany is high due to a network of smaller hospitals with potential over-utilization to offset the higher costs of running a small hospital, see Appendix A.

Australia has a moderately higher number of available beds because there is a network of small rural hospitals, and most elective surgery is conducted in smaller, privately owned hospitals, which flex their staffing to reflect the surgical workflow rather than staffing the beds per se. The region highlighted as ‘medium’ in Figure 1 is probably close to the optimum number of available beds, which this study seeks to identify. This number is then interpreted from patterns of operation and hospital size as per Australia, as well as the degree of implementation of helpful policies such as integrated care, etc.

By way of comparison, countries with similar bed provision to England are: (deaths per 1000 population in brackets), Fiji (8.0), Equatorial Guinea (8.5), Sweden (8.6), Kiribati (6.3), Colombia (5.4), Comoros (8.2), and Uruguay (10.1). All these countries roughly lie along a line of bed equivalence, which is 28% below a line with an intercept = 700, or the middle of the ‘medium’ bed number region. All these countries lie along this line for vastly different reasons. None of these countries other than Sweden can lay claim to a highly effective health care system. Sweden invests heavily in outpatient and long-term care, has a higher number of doctors and nurses per person [31], and has a long history of policy investment in integrated care [32]. The other countries have lower bed numbers, usually due to a lack of funding.

It is now useful to investigate if the widely used metric of beds per 100,000 population can also be modified along similar lines.

### 4.2. Beds per 1000 Deaths Compared to Beds per 1000 Population

For any analysis of bed numbers to be successful, there must be some adjustment to reflect the population age profile, and deaths per 1000 population is the measure that is most readily available. To this end, Figure 2 shows the same data as in Figure 1 but using the ratio of beds per 1000 population versus deaths per 1000 population.

To investigate the relationship between beds per 100,000 population and the crude mortality rate, the groups used in Figure 1 (poorest through very high) were recalculated from beds per 1000 deaths to beds per 100,000 population. Several countries then appeared as outliers and were moved between groups. A range of functions available in the Microsoft Excel curve fit application were tested, with a power law function giving the highest R-squared. The constant ‘A’ and power function ‘B’ for the power law function were then determined by a curve fit for each of the groups (poor through very high).
Beds per 100,000 population = A × (Deaths per 100,000 population) ^B^

The values A and B are likewise related, such that the power function B equals:B = 0.2168 × ln(A) − 0.3042 (R^2^ = 0.9864)

This then leads to a set of lines of equivalence defined by the intercept (the value of A) when deaths per 1000 population are equal to 1, which is the minimum international value for the crude mortality rate.

Hence, the two methods are approximately comparable. It must be pointed out that the method based on beds per 100,000 population gives slightly less discrimination between countries and, in addition, has no theoretical basis as has been established for the role of deaths. The new method is therefore superior to beds per 100,000 population, even after adjustment for the role of population age structure as approximated by the crude mortality rate.

The method will now be employed to study trends in bed supply between countries.

## 5. Trends in Bed Supply

For the method to be valid, it must be able to interpret the trends in bed supply over time. To this end, Figure 3 shows a 20-year trend in bed supply (2000–2019) for 8 randomly chosen European countries. By way of comparison, a 30-year time series has been added for England.

Firstly, note that the crude mortality rate in all countries is changing with time. Declining in Türkiye through rising in Germany. If the method is valid, in the absence of changes in bed supply, each country should lie parallel to a line of equivalence. Countries with the fewest deaths will show higher levels of zig-zag movement depending on year-to-year changes, as per Figure 1. The following comments apply:

Germany commences with a very high bed supply, which is rapidly reduced during the first six years. The reduction continues thereafter, but at a slower pace. The rate of reduction in bed supply appears to be slowing, and the most recent trend is decaying to approach a slope that is closer to a line of equivalence in an aging population (higher crude mortality).

Ireland commences with a high apparent bed supply and lies parallel to a line of equivalence for the first nine years. The sudden reduction in 2009 is due to a change in counting in which long-term-care beds are more correctly classified as ‘nursing home’ beds. Thereafter, bed supply remains roughly unchanged.

Türkiye commences with very low bed supply in 2000 and then rapidly expands bed provision. By 2017, this expansion ceases, and supply lies parallel to a line of equivalence.

Denmark commences with a higher bed supply in 2000 and reduces bed numbers. From 2016 onward, bed supply remains constant and lies parallel to a line of equivalence.

Romania commences with a higher bed supply, which is then reduced. From 2010 onward, supply remains constant and lies parallel to a line of equivalence.

Bulgaria shows no obvious changes in bed supply and lies roughly parallel to a line of equivalence. This country looks to be subject to higher volatility in year-to-year crude mortality, suggesting higher volatility in infectious outbreaks.

Belgium commences with a higher bed supply, which is continuously reduced through 2019. Belgium experiences the smallest change in crude mortality during the 20-year period, with a slight trend to lower crude mortality over time.

Czechia commences with a higher bed supply, with a slight reduction through 2007. In this period, the line is roughly parallel to a line of equivalence. Beyond 2007, bed supply is rapidly reduced. Czechia shows a U-shaped change in crude mortality, reaching a minimum around 2008.

England began in 1990 with higher beds, which were rapidly reduced through 2000. A scheme to increase bed supply then culminated in 2004, after which beds slowly declined but showed a dramatic decline due to measures to correct an NHS deficit implemented in 2010, which arose from an unfortunate coincidence of higher-than-average deaths in 2007 and 2008 and in response to austerity measures imposed on the NHS following the earlier 2008 international financial crash. The crude mortality rate declined from 1990 to 2011, after which it began to increase.

Hence, the method does indeed appear to follow a line of equivalence under conditions where bed supply seems to be roughly adjusted to reflect the prevailing deaths. Changes in the definition of a curative bed are relatively easy to discern, as is the dramatic shift seen in Ireland.

The consequences of higher deaths in 2007 and 2008 upon health care financial pressures (the morbidity-mortality relationship) plus imposed financial austerity from 2010 onwards (from the 2008 financial crash) are shown in England as hospitals attempt to save costs by cutting bed numbers.

It is appropriate to turn our attention to the factors that may influence the supply of beds in different countries.

## 6. Wealth and Bed Supply

The availability of government funding for healthcare will have a profound effect on hospital bed numbers, and Figure 4 demonstrates this relationship using gross national income (GNI) per capita as a measure of wealth [33]. GNI has been used since it is a readily available international metric.

As can be seen in Figure 4, below $10,000 per capita, there is a strong relationship between bed supply and GNI. The most notable exception is turbulent Venezuela, where a GNI of $13,000 is associated with a relative number of beds that is 67% below what is expected. This level of GNI should be associated with bed supply close to the 0% relative difference reference line.

The group of countries with GNI below $10,000 but high relative beds is of interest, and inspection reveals that these are all former or current Soviet bloc countries. This issue is explored in Figure 5, where an inherited Soviet-style centrally planned health care system can be seen to be overly reliant on hospital beds. Many of the former Soviet bloc countries gained independence around 30 years ago, and after 30 years of reform and higher GNI, they are beginning to approach levels of bed availability seen in non-Soviet countries.

However, above a GNI of $20,000 per capita, there is still gross divergence between countries, indicating that different health care systems run at very different levels of bed supply.

This study will now attempt to extend this new method to define a potential optimum region for different types of clinical need, i.e., acute care, medical care, mental health, longer-term (geriatric) care, etc., and to make a comparison of the levels of nurses, doctors, and surgeons.

## 7. Defining an Optimum Region

### 7.1. Medical and Critical Care Beds

A likely optimum region for adult critical care beds has been recently investigated, see Appendix A. It was concluded that in the absence of futile end-of-life interventions, this optimum was likely to lie in the region for occupied beds between a line of equivalence with an intercept of 60 down to somewhere above 30. This can then be turned into available beds based on the size of the CCU. This region encompasses the available CCU beds in the wealthiest Indian states and is at the lower edge for US states. The level of futile intervention in the USA is judged to be high, see Appendix A, hence all but nine US states lie above the line of equivalence with an intercept of 60. All but a few wealthy Indian states lie below a line with an intercept of 60.

The issue regarding medical beds was recently addressed, see Appendix A. A likely optimum region may lie below the line of equivalence with an intercept of 450 and above a line with an intercept of 375. Occupied medical beds in England for the 21 years between 1998/1999 and 2019/2020 lie along the line of equivalence with an intercept of 375, see Appendix A. Since it is known that England has too few medical beds (relative to current demand), the optimum region will have its lowest limit above an intercept of 375.

It goes without saying that the demand for medical beds will depend on the mix of pathogens afflicting the population each year (see Section 11). In this respect, the level of occupied beds in England can reach as high as a line of equivalence with an intercept of 420; hence, below 420 only copes with years when the pathogen mix is less malignant in its actions. This will be discussed in Section 11.

### 7.2. Psychiatric Beds

The required number of psychiatric inpatient beds is a highly debated topic. In one study, an international board of advisors nominated some 65 Delphi panel members. Sixty psychiatric beds per 100,000 population were considered optimal and 30 the minimum, while 25–30 was regarded as mild, 15–25 as moderate, and less than 15 as severe shortage [34]. As an approximate comparison, 60 beds per 100,000 population, or around 8.8 deaths per 1000 population, is equivalent to 70 beds per 1000 deaths. Between 1990 and 2000, the median number of psychiatric beds in less developed countries decreased from 3.0 to 2.2 per 100,000 people (median percentage change = −16.1%). Beds in forensic and residential facilities are nonexistent in most less developed countries [35]. Figure 6 shows the availability of Psychiatric beds for European countries in 2019.

The figure of 60 beds per 100,000 people mentioned above could feasibly lie between lines of equivalence with intercepts of 100 and 200. However, somewhere below an intercept of 100 looks possible.

In 1978, Italy implemented an aggressive shift from hospital-based to community-based care. Compulsory admissions dropped from 20,300 in 1978 to around 8800 in 2015 [36]. Psychiatric beds fell from 160 beds per 100,000 population in 1978 to less than 20 from 1999 onward [36]. Hence, it is possible to operate with fewer psychiatric beds, provided that suitable levels of outpatient/community care are in place. However, note that this transition in Italy took about 20 years to fully implement and would require substantial investment in psychiatric community staff [36]. This probably explains why all other countries lie above the line of equivalence with an intercept of 100.

A study in the USA calculated that at optimum levels of community mental health (CMH) workers, there are 35 psychiatric beds per 100,000 population, equivalent to 42 beds per 1000 deaths [37]. This is higher than the UK and contains the inbuilt assumption of optimum levels of CMH workers.

Note that the occupancy levels in English psychiatric hospitals rose from around 86.5% in 2011 to above 89.5% from 2015 onward [19]. Given that the psychiatric bed pool encompasses everything from secure units to adolescent units, etc., the effect of size on the occupancy rate (discussed in Section 10) means that somewhere around 90% represents the upper possible limit for average bed occupancy given that the total is made up of smaller specialist and general units. In general psychiatric units, occupancy is often at 100%, and patients are often moved to an available space that is many miles from their home. This implies that the level of outpatient/community support in England is inadequate relative to the available inpatient beds. The shift from an inpatient-based model has been hampered by under-investment in outpatient/community staffing, the importance of which was emphasized in Italy.

### 7.3. Acute Rehabilitation Beds

Data on the level of acute rehabilitation beds in European countries are presented in Figure 7.

The usual wide difference between countries is evident. Data for the lowest country (England) only includes actual occupied bed numbers for specialty Rehabilitation, although bed occupancy will be in the high 90% region. The issue is complicated by the fact that rehabilitation beds for army veterans are often charitably funded, and rehabilitation units can be in non-acute and private hospitals. It looks like an optimum region could exist lying between lines of equivalence with an intercept of 100 to 200. Further analysis is required on this topic to include a count of all rehabilitation beds, irrespective of their location inside or outside of an acute hospital. As always, this will rely on suitable investment in outpatient/community support.

### 7.4. Acute Long-Term Care Beds

The situation for European countries is given in Figure 8.

An optimum lying between lines of equivalence with an intercept of 100 to 200 looks possible. Data for England are the lowest data value along the intercept = 100 line. The England data have been taken from the inpatient specialty “Geriatric care”, which is delivered by consultant geriatricians. Both geriatricians and geriatric beds are in short supply relative to the demand. Patients requiring geriatric care can therefore be found in general medical beds. Impatient provision lower than an intercept of 100 may be possible with suitable investment in outpatient/community staff.

### 7.5. Physicians and Nurses

The positions regarding Nurses, Doctors, and Surgeons are given in Figure 9. The data have been scaled to fit with the lines of equivalence, which are there merely as a reference point. Sweden is identified as a triangle. There are generally more nurses than doctors. The data for surgeons appears to show gross variation, which may be partly due to the interpretation of the definition. The most striking outcome is that surgeons in the UK appear to have one of the highest levels in the world, matched only by Greece. Overprovision appears likely—subject to validation regarding interpretation of the definition.

The provision of nurses in the UK is on the low side and is equivalent to that in Brazil, Czechia, Lithuania, the Maldives, and Russia. The provision of doctors is also on the low side, with the UK equivalent to Armenia, Antigua and Barbuda, Belgium, Chile, El Salvador, Malta, and Ukraine.

Compared to the UK, Sweden has 40% more nurses, 54% more doctors, but 14% fewer surgeons.

Appendix A [3,41] provides further analysis using occupied rather than available beds in England. This covers a random selection of specialties. The overall conclusion is that the new method works well, provided that the historic context behind the data is considered, i.e., increasing sub-specialization within general medicine and surgery, the impact of austerity following the financial crash, etc.

As I have observed over the past 30 years, English politicians have made totally unsubstantiated claims that the NHS has too many beds, and Figure 1 shows that the lowest level of beds for a developed country has been achieved without the necessary investment in nurses and doctors to support such a low number of hospital beds. The UK invests around 20% less of the share of GDP into healthcare than Sweden [42], yet it expects to be able to operate with fewer beds. It should be noted that England managed to go through the COVID-19 pandemic simply because vast amounts of ongoing surgical and medical care were shut down [43], which then shifts the expressed demand to a future date, leading to one crisis followed by another.

## 8. Adjusting International Data for Relative Population Deprivation

It has been noted that adjustments need to be made for some measure of ‘deprivation’. The study relating to English Clinical Commissioning groups used the UK Index of Multiple Deprivation (IMD), see Appendix A, while a study in Australia used the proportion of state population that were indigenous people, in Appendix A. Neither of these is applicable in an international context. One readily available indicator is the age-standardized mortality rate (ASMR), which can be obtained from the WHO and other sources [44].

Appendix A [43,44,45,46,47,48,49,50,51] and Appendix A [52,53,54,55,56] provide wider analysis supporting the notion that ASMR is an appropriate international and local variable for the effect of ‘deprivation’, and discussion on the possible range in the value of the slope for total hospital beds versus ASMR.

Appendix A in the Appendix A shows the range in ASMR for world countries and UK local authorities with different values of deaths per 1000 population [41,44]. As a generalization, ASMR tends to increase as the raw mortality rate increases; however, the key point is that at any value of the crude mortality rate, there are a wide range of values for ASMR.

As a reference point, Appendix A shows the ASMR for the 50 countries with the lowest value for ASMR. Sweden is shown as the reference point for the lowest number of total curative hospital beds in a developed country. The key observation from Appendix A is that England and the UK have 14% and 17%, respectively, higher ASMR than Sweden, which suggests that the higher relative deprivation/lifestyle factors cannot justify the lower level of total curative beds noted in Figure 1. The suggestion is that England and the UK have reached this point simply by restricting hospital bed numbers without supporting investment in outpatient/primary care, as has progressed over many years in Sweden. Hence the disruptively high bed occupancy in England, to be shown later.

If we assume that a higher ASMR is a measure of deprivation or hardship, then countries with a higher ASMR would have a higher intrinsic demand for hospital beds, which would go unmet in the poorer countries. Appendix A investigates the likely slope of the relationship between total available hospital beds and ASMR. The slope is estimated to lie somewhere in the region of 13 to 34 higher total hospital beds per 1000 deaths. Finally, Appendix A investigates the discrepancy between bed supply and demand in US states. It is concluded that wealth and not need drive bed supply in the USA, as has been extensively reported by others [53,54,55]. Having investigated the relevant factors regarding bed supply and the use of the new model, we can now turn to the topic of applying the new model at the local level.

## 9. Defining Optimum Bed Supply at the Local Level

All models contain hidden assumptions, necessitating comparison between models. It is emphasized that this method should be used alongside other methods when attempting to specify the exact size for a local hospital [56,57,58,59,60,61,62,63].

Indeed, all models based on an annual average will underestimate true bed demand simply due to the seasonal nature of demand [64,65] and Appendix A. Additionally, surge capacity is required to allow beds to be taken offline for a deep-clean and to cope with pandemics and catastrophes [63].

The following sections will provide a framework for applying the international method at regional and local levels.

### 9.1. Data Availability at the Local Level

All developed countries have a system of collecting data on deaths, births, and population at regional, local, and small-area levels. The smallest area for such aggregation is usually called a ‘census tract’ in the USA or, in the UK, an ‘output area’. In the UK, an output area contains roughly 300 people, while a census tract in the US contains about 4000 people. Each census-sized area will have a population centroid (Eastings and Northings), which is essentially a grid reference (Cartesian coordinates).

Either a Geographic Information System (GIS) using travel time, or a simple spreadsheet calculation (using straight line distance) can then be used to allocate each census area into the catchment area of a hospital, see Appendix A. Manual adjustment may be necessary when rivers or mountains prevent access. The relevant data can then be aggregated to give the effective deaths per 1000 population or beds per 1000 deaths for that hospital.

A sensitivity analysis should be conducted for the inclusion/exclusion of marginal census areas; however, a more pressing issue relates to forecasting future trends.

### 9.2. The Crude Mortality Rate at Sub-National Level

In all developed countries, the crude mortality rate is available at the small-area level. This is important when attempting to take the model to a local level. As an illustration, Figure 10 shows the crude mortality rate for a selection of English and Welsh local authorities in 2019 (the pre-COVID-19 year).

While the value of the crude mortality rate for the UK was around 8.8 deaths per 1000 population in 2019, the range at the local authority level is from around 3 to 15, which roughly spans the international range shown in Figure 1. In the UK, low values of crude mortality only occur in inner-city locations.

From an international perspective, values of crude mortality higher than 13 deaths per 1000 population only occur in the former Soviet countries such as Moldova, Croatia, Hungary, Ukraine, Belarus, Serbia, Lithuania, Latvia, and Bulgaria (16.3 deaths per 1000 population). Such high levels of the crude mortality rate are only approached in a minority of “elderly” communities in the UK. Clearly, values of crude mortality above 15 deaths per 1000 population will occur at the small area level, especially in areas with high numbers of nursing home beds.

Hence, the crude mortality rate is a widely available measure that enables discrimination between communities and their associated bed demand at the local level.

The same applies for ASMR, where country specific ASMR calculations based on a relevant national population should be used rather than a world population, etc. Regarding ASMR, it should be noted that the density of nursing home beds has a skewing effect [66,67], as do healthcare behaviors in urban and rural settings [68,69], see Appendix A.

### 9.3. Effect of the World War II Baby Boom on Future Deaths

The cessation of WWII led to a baby boom in most participating countries. In addition, countries such as the USA and Australia initiated programs for inward immigration of the many displaced persons arising from the war. The leading edge of the baby boom turned 65 in 2011—65 being the age at which death occurs at increasing frequency. Such countries will all be experiencing rising deaths over the next 30 years [70]. That part of bed demand relating to changes in deaths has therefore been activated. In addition, trends in births and migration will imply that each country will have its own unique profile of future deaths. For example, the Russian Federation has a declining population arising from the loss of young men during the 1979–1989 Afghanistan war [71].

Such future trends are incorporated into the new model via the use of deaths per 1000 population in the x-axis, as was illustrated in Figure 1. The effects of the WWII baby boom are then incorporated into national and local forecasts of both population and deaths.

### 9.4. Forecasting Future Deaths and Population

All methods for bed planning rely on forecasting the future value of the model inputs. It is the authors’ experience that even national-level forecasts of deaths are highly unreliable due to the difficulty of defining the base year for the forecast, in Appendix A.

Defining a base year is problematic simply because deaths are so volatile, as witnessed in Section 11. To address this problem with the base year, a method based on 11 years of previous data has been proposed, in Appendix A. The issue becomes more problematic at the local level, and multiple scenario methods along with uncertainty intervals must be employed.

The aim of such scenarios is not to derive the minimum possible answer but to construct the likely range for the future. Figure 11 illustrates how such a baseline trend can be constructed, in Appendix A. A second-order polynomial has been used to estimate the baseline trend, which intersects the points of actual minimum deaths (the dark blue line). The COVID-19 pandemic illustrates a period of unanticipated “shock” demand that was far larger than similar ‘shock’ increases experienced prior to this pandemic.

Figure 11 uses data for the whole of the USA, but the method can be replicated in states and smaller geographies, where local issues become more influential. Note the existence of points for minimum mortality as for the 12-months ending 7 December, 8 December to 10 December, etc. Also note the existence of a period of higher deaths lying roughly between 14 December and 19 December. This period of higher deaths is also seen in the UK, in Appendix A.

All forecasts must reflect the reality that deaths and associated demand are highly volatile, with direct implications for the hospital bed occupancy margin in Section 10.

## 10. Defining an Optimum Hospital Size

### 10.1. Average Bed Occupancy as a Measure of Operational Chaos

Hospital bed occupancy has been traditionally measured at midnight, and for international comparison, we use an annual average. There are several problems with this approach because:

The average occupancy rate shows seasonal changes, with the highest occupancy during the winter months (rainy season at the equator) due to the higher incidence of winter infectious diseases and the effects of cold [64,65]. Other cycles operate nearer to the equator, usually based on the rainy season.

Occupancy also has a daily cycle, with a minimum around midnight and a maximum during the day.

Queuing theory clearly shows that the likelihood of a patient being ‘turned away’ due to no bed being available (hence a measure of chaos, inefficiency, delays to admission, and waiting lists) increases as the real or instantaneous occupancy increases [72,73], see Appendix A. Smaller hospitals or bed pools must operate at lower average occupancy due to the higher volatility in admissions as size decreases. Hence lower average occupancy in pediatric, maternity, and critical care bed pools and in countries with smaller hospitals [72,73], see Appendix A.

With these limitations aside, rising average midnight occupancy is an excellent measure that there are insufficient beds relative to the expressed demand. To this end, Figure 12 shows the trend in annual average midnight occupancy for curative beds in England from 2010 to 2019.

While 85% average occupancy is often quoted as the optimum for hospital occupancy, there is no theoretical basis for this claim [72,73], and in Appendix A. An annual average rate of 85% does, however, roughly apply to very large hospitals with more than 1000 beds, see Appendix A. The reason that 85% is so often claimed to apply is that it also represents a measure of perceived busyness, see Appendix A, which would be independent of size.

Since 2010, financial austerity in the English NHS has led to the closure of beds to reduce costs. Figure 12 is therefore the outcome of such imposed austerity. Note that the increase in average occupancy slows with time simply because it becomes increasingly difficult during the daytime to admit a patient, leading to queues for admission and cancelled elective operations. Note that over this time the national waiting list for an elective operation grew by millions of people [74], and widespread curtailment of elective surgery regularly occurs during the winter months (further fueling growth in the waiting list), and there are typically queues of ambulances waiting outside hospitals during the winter [75]. English hospitals have simply descended into unfunctional chaos, even though they are generally very large by international standards. This is a by-product of high population density, and the government attempts to minimize costs by limiting the number of hospitals. For example, small US towns will typically have two or more hospitals due to private healthcare company competition and a network of small community hospitals throughout the extensive rural low-density US population [76].

Each English hospital will have a large “bed management” team attempting to squeeze patients into any available bed—called outlying patients—who typically have a higher average length of stay; receive poorer quality care; and have a higher readmission rate [77,78,79,80]. The mortality rate may not be greatly affected, although adjustment for risk factors is difficult [77,78,79,80].

By way of comparison, in 26 European countries with data available for 2019 [29,30], the median occupancy was 73% (69% lower quartile, 79% upper quartile). The minimum average occupancy was 62% in Cyprus (a small country), up to 90% in Ireland. Germany had an average of 79%. Other countries lying in the range of 78% to 79% were France, Italy, Luxembourg, and Liechtenstein. Ireland has recognized that occupancy is high due to a deficit in beds, with plans to expand bed numbers [59]. Occupancy in England is seemingly the highest in Europe. High bed occupancy has consistently been associated with every possible measure of (imposed) hospital inefficiency [80,81,82,83,84,85], see Appendix A.

As a final note, beds per se do not make a large contribution to costs; it is the staff associated with the beds that is ‘expensive’.

### 10.2. Optimum Hospital Size as an Economy of Scale

While the average bed occupancy rate gives insight into the level of operational chaos, there is also the related issue of the optimum size for technical efficiency. Economy and diseconomy of scale issues were explored in the review of Giancotti et al. [86], which gave evidence for economies of scale in hospitals with 200–300 beds. Diseconomies of scale occur below 200 beds and above 600 beds [86]. Queuing theory shows that larger hospitals can operate at higher average occupancy [72,73], see Appendix A, and as a hospital progresses beyond 600 beds, it seems that increasing specialization leads to higher costs per patient. The diseconomy of scale below 200 beds is due to the lower average occupancy associated with small sizes, leading to higher capital costs and higher staffing costs per patient, see Appendix A.

### 10.3. Length of Stay and an Efficient Hospital

During the 1970s and 1990s, hospital length of stay (LOS) experienced a rapid decline, see Appendix A. This was mainly due to the simple discovery that rapid mobilization was far more effective than bed rest for recovery after injury and surgery [87,88,89]. Had such a rapid reduction continued, the net LOS would have declined to zero well before the 2020s. Hence, any model for LOS must incorporate a decline in the rate of reduction in LOS over time [60,61]. This effect is illustrated in Appendix A.

LOS does depend on age; however, one study on LOS in the adult medical intensive care unit showed that LOS was independent of age for patients who survived but depended strongly on age for those who died [90]. This observation could suggest that hospitals that service a population with a high crude mortality rate will experience greater pressure from high LOS.

One potential effect of the aging population on LOS is that LOS may well reach an asymptote and then start to increase as the proportion of older people increases. I have yet to see any models for hospital bed numbers that incorporate such a possible outcome. See Appendix A.

The study by Hughes et al. [77] shows that LOS “outliers” make a significant contribution to the average LOS and that such “outliers” should not be excluded from the LOS calculation or from the number of occupied bed days. Rachoin et al. [91] observed that medical patients with a higher LOS tended to have higher readmission rates. Indeed, the NTD/TTD phenomenon discussed earlier implies that patients in the last six months of life will be characterized by high rates of readmission—an association only realized in hindsight.

The study of Walsh et al. [92] observed a strong relationship between constrained bed supply and length of stay. Between 2010 and 2012, while length of stay fell by 6.4%, approximately 42% of this reduction was associated with the decline in bed supply. The use of length of stay as an efficiency measure should be understood on the contextual basis of other health system changes. A shorter length of stay may be indicative of a lack of resources or available bed supply, as opposed to reduced demand for care or the shifting of care to other settings [92].

In the English NHS, the headline LOS has been declining, but closer inspection reveals that this is largely due to the inclusion of millions of zero-day stay emergency admissions due to “admission” of persons into short-stay emergency assessment units. I would claim that the majority of such “admissions” are more correctly defined as emergency department attendances. The reality was that many such “admissions” were simply an attempt to circumvent the emergency department’s four-hour target introduced by the Blair government in 2004.

As an example, the real LOS for Shigellosis (A03) is around 5 days and is only declining at around 0.03 days per annum (0.6% p.a.), while that for Amoebiasis (A06) is around 11 days and is declining at around 0.1 days per annum (0.9% p.a.) [3]. The reality is that the residual genuine overnight stay LOS is not declining at any great pace or is staying roughly unchanged, in Appendix A. See Appendix A for an example.

This does not imply that emergency assessment units should not be opened, but rather that the planning for such units should incorporate separate estimates of the real-time LOS in such departments and the likely future growth in admissions. Assigning a zero LOS to any person admitted and discharged before midnight is simply an obfuscation of reality and an open door to underestimate real future bed demand. All hospital planning must be conducted based on reality and not on wildly optimistic ‘concocted’ estimates of future LOS.

While it is true that poorly executed hospital processes and procedures of care can lead to a longer LOS, it has been my observation that the processes of clinical coding, which include the training of doctors on the nuances of recording patient information, often lead to spurious examples of long LOS. All hospitals are well advised to ensure that all available information is collected regarding every in-hospital death, i.e., blood biochemistry and other diagnostic test results, any imaging studies, etc. It is a fundamental risk to rely on some hastily written diagnosis by a junior doctor given that most deaths involve a greater complexity of multimorbidity and other factors, all of which should be reflected in an extensive series of clinical codes for each patient. Some hospitals have an evident lack of detailed coding, which then distorts the interpretation of LOS.

It is also far too easy for a Management Consultancy to construct a plausible story (on paper) as to why future LOS will be reduced, and hospital Executives and senior Managers will be unaware of the hidden pitfalls. The series of articles under the title “Understanding Length of Stay” in Appendix A should be studied, see Appendix A.

### 10.4. Staffing Regulates Costs Not the Physical Number of Beds

It has been the author’s observation over many years that modeling of future bed demand in the UK appears to be stricken by the fallacy that the number of beds is the primary driver of hospital costs. The capital cost of a bed plus floorspace is covered under capital costs and is amortized yearly over the expected lifespan of the building or the physical bed. The annual cost of additional beds per se is therefore surprisingly low. The initial sections of this study clearly showed that the demand for the beds is highly volatile, and this implies that the staffing of the beds needs to be highly flexible. The actual cost driver is the number of staff as full-time equivalents. A simple method is available to optimize the mix of full-time and flexible staffing requirements, in SL.31, but it is never used.

My discussions with UK NHS Directors of Personnel are that flexible staffing is viewed as an “impossible” task. Clearly, what is imperative to achieving minimum operating costs is viewed as a “no-go” area and is not reflected in overall NHS policies for NHS staffing. Part of the problem is the very high bed occupancy levels that now afflict the UK NHS. At high levels of bed occupancy, the illusion is created that the beds are always fully staffed; hence, the number of beds becomes a perceived cost and not a route to efficiency.

### 10.5. Delayed Transfers of Care (DTOC)

DTOCs arise at the interface between hospital and social care and can be reduced by local partnerships and discharge planning [93]. They create an increased length of stay. However, in England, they are often presented as a major reason why available beds appear too low compared to the expressed demand. The problem is that when you are in a crisis, all manner of superfluous reasons are given as the “cause” of the problem.

Over a 30-year period, it has also been my observation that DTOCs rise and fall in proportion to available funding, which is influenced by the financial pressures arising from unrecognized outbreaks of infectious agents, see Appendix A. Any agent that pushes a vulnerable proportion of the population toward death will simultaneously increase hospital admissions and the pressure on primary care, community nursing, and social care. All parties will simultaneously experience higher demand/costs, which will inevitably lead to subtle causes of DTOCs. On some occasions, such insights can only be gained from experience gained over many years. This is discussed in detail in Section 11.

DTOCs will never drop to zero, and the long-term fluctuations in the level of DTOCs simply become part of the reserve or surge capacity required to run an efficient hospital. The next section is highly relevant to DTOCs and wider aspects of explaining why bed demand is so uncertain.

## 11. Pathogen Outbreaks, Bed Demand, and the Bed Occupancy Margin

The logic behind this line of research lay in my early observations that activities associated with hospital demand, such as all types of outpatient, inpatient, and emergency department attendances/admissions, showed year-to-year variation that was between 2- and 3-times higher than could arise from Poisson variation, see Appendix A. The same can be said for the length of stay, in Appendix A, deaths, in Appendix A, and healthcare budgets, in Appendix A. Indeed, admissions for some diagnoses and certain types of cancer show extreme volatility, indicating high sensitivity to the external environment, in Appendix A. Appendix A illustrates the issues regarding volatility for deaths and hospital bed demand. Such high levels of inherent volatility demand an explanation.

### 11.1. Death as a Proxy for Wider Admissions

For individual pathogens, it is possible to estimate the number of infections from the reported deaths [94]. The action of infections on acute demand can be very subtle. For example, people feeling unwell due to an infection may experience a higher number of injuries, falls, and fractures, most of which may not immediately lead to death. However, in other people, the same infection may exacerbate other conditions, which may involve death. If we accept the proposal that deaths are serving as a proxy for acute admissions, Figure 2 shows the trend in overnight stay emergency admissions per death in England from the financial year 1998/1999 through to 2019/2020 (the last financial year before the COVID-19 pandemic). Excluded from this analysis are all elective admissions, any same-day stay admissions, and any admissions relating to Maternity, Pediatrics, Community Care, or Mental Health, i.e., acute overnight-stay hospital adult care only. 

As can be seen in Figure 13, the number of such overnight stay (adult) admissions per death appears to range from around 4.5 before 1998/1999 to 7.5 in 2019/2020. It should be noted that deaths in England reached their minimum number between the 2009/2010 and 2011/2012 financial years [95]. The general increase in this ratio from 1998/1999 onward may reflect an expansion in the range of acute medical/surgical interventions that can be offered for emergency hospital care. This range of care roughly expands at 0.12 adult emergency admissions per death per year—although perhaps more so before 2009/2010. It should be noted that admissions for certain mental health conditions are also exacerbated by infections [96]. Indeed, one-third of COVID-19 survivors receive a neurological or mental health diagnosis in the six months following infection [97,98].

Even if it were argued that the real ratio amenable to exacerbation by infections is only, say, 4 admissions per death, the absolute number of deaths is still acting as a proxy of the demand for acute adult emergency care and that this ratio is subject to year-to-year volatility—after correcting for underlying growth in the ratio.

In the field of statistical process control (SPC), a major observation is that ‘the variation is the voice of the process [99]. In terms of hospital bed planning, this translates into something like, “unless you understand why healthcare is so variable, you will never be able to create a credible model of bed demand”. Having discussed the role of unrecognized infectious outbreaks on human health and the consequent need for hospital beds, we can now consider how the absolute number of deaths can be incorporated into hospital bed modeling.

### 11.2. Evidence for Such Outbreaks

From Appendix A, age-based forecasting only ever delivers slightly curved continuous lines with a low slope. Yet the real-world of hospital admissions and costs shows very high volatility, in Appendix A, which includes high volatility in winter deaths due to infectious outbreaks, in Appendix A. Hence, the reader can view some 30 years of research publications as an attempt to introduce some understanding to the question, why?

One of the more interesting observations from my studies was that the trends in emergency admissions appeared to be behaving like a series of infectious outbreaks over and above the usual winter influenza outbreaks, see Appendix A. A series of studies, in Appendix A established that both deaths and hospital admissions (especially emergency medical admissions) showed high levels of small-area spatiotemporal behavior, or spatial granularity. Such spatiotemporal behavior is a characteristic of infectious spread [100,101,102]. Also, some diagnoses were affected while others were not, in Appendix A.

Interesting confirmation of such a possibility comes from the fact that, as of 2022, there were around 3000 species of known human pathogens, in Appendix A. The large majority are unexplored regarding their clinical effects, and indeed, no one routinely tests for their presence—other than the more commonly known species. A study on the human DNA virome, i.e., persistent viruses, using the 17 most common persistent DNA viruses showed that individuals were infected with between 2 and 17 of these 17 most common persistent DNA viruses [103]. Individuals were most infected with 6–7 of these viruses in 39% of cases. At the sub-species level for each virus, there was profound genetic variation between individuals. Table 1 demonstrates the distribution of these 17 common persistent DNA viruses in human tissues.

The top line gives a count of how many DNA viruses were found in each tissue. The percent value in bold is for the most common virus in that tissue. The mechanisms behind how such persistent and other infections may interact with human health are covered in Section 11.3. Interpret this table as indicating the potential to influence health in each organ/tissue and that DNA viruses are only half the full picture.

Modern high-speed/volume travel (air/rail/road) implies that all pathogens (persistent or occasional) are spread rapidly around the world [104,105,106]. As if 3000 species of pathogen were not of sufficient concern, each species represents a wide range of sub-species, strains, and variants with a diverse range of clinical effects—the study of which is termed metagenomics [107,108,109]. Also, dual infections (superinfection and coinfection) are very common [110,111].

Is there any evidence that unknown pathogen outbreaks are a frequent occurrence? The best evidence for such a proposal comes from the small-area behavior of deaths, which are displaying apparently ‘unexplained’ short-term trends, see Appendix A. Hence, for each area, there will be a general long-term trajectory for deaths that depends on past births, changes in life expectancy, and net migration. This trajectory will be available in all developed countries and forms part of the population forecasting process. However, the shape of this calculated trajectory will be a smooth line.

Figure 14 illustrates that the actual trajectory is far from a smooth line. The UK local authorities in Figure 14 were randomly chosen with the criteria of having just over 2000 deaths per year in 2001.

The displayed trajectory is a moving 12-month total commencing on 1 December which is relative to the median value of deaths in each local authority. In a moving 12-month total, the usual level of seasonal cycles is removed since every point includes all 12-months. The existence of such trajectories has been largely ignored by epidemiologists because there is no obvious explanation for the observed behavior. Winter cold-snaps or summer heatwaves cannot explain the behavior since they are usually of such short duration as to make a negligible impact on the 12-month total. By choosing local authorities with above 2000 deaths per year, the role of Poisson-based randomness is likewise excluded since ±1 standard deviation of Poisson variation only accounts for ±2% variation. This is reflected in the very small background “wriggles” in the trendlines.

In a moving 12-month total, a spike event such as an influenza outbreak would be expected to generate a “tabletop”-shaped feature. The excess deaths move into the 12-month total, stay in the total for 12 months, and then exit the total. Such a spike event can be seen in the Highlands, commencing in December 2006. Such tabletop features are somewhat less than expected, and the variety of shapes indicates a more complex set of contributory factors. As has been stated, there is no such thing as a single pathogen winter, see Appendix A.

In a moving 12-month chart, a pyramid-shaped pattern is indicative of on/off switching. I have proposed that at a national level, the net contribution from all pathogens and their interactions via pathogen interference is to produce a set of on/off switches in human health. The on-part of the cycle (the upward face of the pattern) initiates roughly 12-months of poor human health, while the off-part of the cycle (the downward face) represents roughly a 12-month period of ‘better’ human health. This cycle had a longer wavelength back in the 1980s, but the frequency gradually contracted to the current 2-year pattern seen since the 2010s, see Appendix A.

While other explanations may be possible, the pattern is real, can be followed over many years, and does indeed affect different medical and psychiatric diagnoses [3].

To explain the residual peaks and troughs, it is illustrative to point to the disruption caused by the COVID-19 pandemic, where deaths peak in the 12-month total ending around March 2021. This is the outcome of a very real pathogen. However, notice the vast difference in COVID-19 deaths between Enfield (London) and Walsall (Midlands) and the almost minor effect in Aberdeen City (Scotland) or Gateshead (N.E. England). For example, note the very high deaths for the 12-month total in 2005 in Rochdale, while Highland experienced a minimum. The highly granular effect in Figure 1 arises from the role of super-spreaders and super-spreader events (SSEs) upon infectious transmission [112,113,114].

Figure A1 shows a similar situation for nine London Boroughs where spatial proximity precludes wider effects due to weather patterns or other regional effects. Hence, in Figure A1, note the low deaths in Richmond-on-Thames during 2009, when the other boroughs are generally high. Sutton is worst affected during an event occurring in 2014/2015, with the highest moving 12-month total measured in March 2015, which barely affects Kensington and Chelsea.

Consider the implications of the high/low points. Hence, a maximum implies an extended period of sustained high deaths, while a minimum implies a switch to low mortality. The low mortality extended periods can be inferred to have a low net pathogen health load and vice versa. Such switching is made possible due to pathogen interference, where infection with one pathogen modifies the frequency and clinical outcome of subsequent infections [115,116], also Appendix A. Pathogen interference arises from the production of interferons, which is regulated by the expression of small noncoding RNAs (miRNAs) [117]—discussed further in the next section. Pathogen interference then amplifies/damages the local effects on mortality, depending on the local mix of pathogens circulating at the point when the new pathogen is introduced via a super-spreader, or SSE.

Differences in timing and magnitude are widely evident in Figure 1 and Figure A1. Spatial analysis of such events shows neighborhood-area behavior characteristic of infectious transmission, see Appendix A.

From Figure 14 and Figure A1, it is concluded that the possibility of previously undocumented (mainly local) infectious events is very real and has a direct effect on total deaths and, hence, on wider morbidity, leading to hospital admission and bed demand. The key point being that the local events show high volatility and, by implication, the need for an appropriate average occupancy in local hospitals to accommodate such fluctuations in demand—indeed; the overwhelming bed demand seen during the early stages of the COVID-19 pandemic. The evidence has seemingly been lying right under our nose, but traditional disease surveillance has completely failed to detect such small-area impacts, see Appendix A.

The central flaw in existing surveillance systems is that they are heavily biased to detect explosive disease outbreaks such as influenza or COVID-19, see Appendix A. This bias is further exacerbated by the methods used to select the baseline for deaths. This problem is amply illustrated in Figure 1 and Figure A1 in that the “true” baseline is hidden among high volatility. One solution is to select a low pathogen baseline that intersects the minimum points in the long-term trend. An example was given for the USA in Figure 11.

At this point, it must be noted that a moving 12-month total does not make any assumptions regarding a baseline and is especially useful at detecting non-standard outbreaks, which may be expected from 3000 species of pathogen, in Appendix A. However, it only works retrospectively.

Current disease surveillance is also heavily focused on a narrow range of pathogens with identifiable symptoms, i.e., Ebola, hemorrhagic fever, influenza, etc., while it is suggested that some 3000 species of pathogen are likely to have a wide range of nuanced effects against pre-existing conditions and will therefore go unreported or, more correctly, misreported. One example of more nuanced associations is the well-known link between influenza and acute cardiovascular events [118].

The issue as to whether the patterns of deaths seen in Figure 14 and Figure A1 are impacting psychiatric bed demand needs to be explored, see Figure 15. As was discussed earlier, infections play an important role in the initiation and exacerbation of mental health conditions. As a proxy for infectious outbreaks, deaths will therefore be one of the explanatory variables in any model for mental health beds. In the UK, the specialty “Old Age Psychiatry” is highly likely to be heavily influenced by nearness to death rather than age per se.

It has already been pointed out that average bed occupancy in acute hospitals in England shows an unexplained pattern of occupancy that is roughly two years wide from trough to trough, in Appendix A, even after calculating a rolling 12-month average—which would remove any seasonal patterns. Figure 14 confirms the same pattern for Mental Illness bed occupancy. Some explanation is required in that the 12-month period from March 2011 onward encompasses a time of rapidly declining bed supply as care for mental illness is transferred into the community.

The curious pattern was observed prior to 2014 but is suppressed due to certain conditions/diagnoses seemingly not strongly affected by the pattern being still treated in an inpatient setting. However, after the loss of 600 beds from 2011 to 2014, the pattern becomes far stronger as the range of conditions undergoing inpatient treatment has been trimmed to a core set of conditions seemingly sensitive to the strange pattern(s).

### 11.3. How Pathogens Interact with Existing Diseases

As per the TTD/NTD phenomena, a study by Hansen et al. [119] has demonstrated that costs in the last five years of life are driven by morbidity and not age. On this occasion, costs imply access to hospitals and other care. How might this be influenced by pathogens? All will be familiar with the function of the immune system during infection; however, only some will be aware that immune function and many diseases are regulated by small noncoding RNAs (miRNAs). The miRNAs act to regulate gene expression, which then impacts metabolic pathways, etc. miRNAs therefore serve as the common interface between pathogens and disease initiation and exacerbation [120,121,122,123,124,125,126]—hence the seeming inexplicable trends seen in Figure 14 and Figure A1.

One expression of this interaction occurs in pathogen interference, whereby infection with one pathogen stimulates the production of a range of miRNAs, which then alter the expression of interferons, which in turn alter the ability of other pathogens to infect that individual [117]. Changes in interferon production are only one of many induced changes in cellular function, metabolism, and immune responses, such as inflammation, which can include damaging cytokine storms [120,121,122,123,124,125,126]. Such processes can then trigger autoimmune conditions and malignancies in genetically and epigenetically susceptible individuals [120,121,122,123,124,125,126].

In such a context, Figure 14 becomes entirely understandable and implies that so-called health service’ demand management’ may be far more nuanced than has been appreciated. The COVID-19 pandemic with associated long-term COVID [97] is an excellent example of the difficulty in ‘managing’ such events given unexpected knock-on effects. Such unexpected knock-on effects are probably more widespread than anticipated and may include poorly understood conditions such as chronic fatigue syndrome or myalgic encephalomyelitis (ME/CFS) and other syndromic conditions, which are surprisingly like long-term COVID.

In my earlier work, the issue was explored as to whether the common persistent human immune modulating virus Cytomegalovirus (HCMV) could influence the range of common diagnoses that changed during periods of unexplained higher deaths, in Appendix A. The answer was in the affirmative. However, in the light of recent developments regarding the human persistent virome [103], the existence of 3000 species of known human pathogens, in Appendix A, and the mechanisms of pathogen interference, in Appendix A, it is probably better to substitute the words ‘any persistent pathogen’ for ‘CMV’, and that infection with any occasional pathogen, i.e., influenza, respiratory syncytial virus, parainfluenza, etc., will trigger a cascade of events leading to the re-activation of some of the persistent human virome. This then triggers a further cascade of consequences.

The most common persistent virus, HHV-6, is implicated in many disease processes and malignancies [127,128,129]. Reactivation of EBV, CMV, and HHV-6 is common in critically ill people with COVID-19 disease [130].

### 11.4. Wider Indicators of Health Are Also Affected

It should be noted that wider indicators of human health, such as worker sickness absence, in Appendix A and General Practitioner (GP) referrals for a specialist opinion, in Appendix A, also appear to be influenced by the same forces that regulate the above fluctuations in deaths. If both occasional and persistent pathogens were exerting wider-than-expected effects on health, such associations are to be expected. The trends in sickness absence are illustrated in Appendix A.

At this point, it should be highlighted that fluctuations in hospital admissions for certain diagnoses also appear to be associated with the same root cause, see Appendix A. Indeed, the unexplained fluctuations in the ratio of male to female deaths, hospital admissions, see Appendix A, and the costs for certain conditions may also fall into this category, in Appendix A.

Figure 16 shows three among many unexplained trends in admissions for primary diagnoses in Chapter R (Signs and symptoms) of the International Classification of Diseases (ICD-10) primary diagnosis.

No one investigates such unexplained trends, and while some of the movement may be due to changes in how symptoms are coded, the more interesting feature is the short-term fluctuations, which cannot be explained by any long-term changes in the process of coding or the definition of an admission. For example, why the sudden spike in admissions for unknown/unspecified causes of morbidity in 2018/2019, for fever of unknown origin in 2010/2011, or the maximum in 2004/2005 for abnormal serum immunology? Further investigation shows changes in the ratio of male to female admissions concealed in these trends—also largely unexplained, in Appendix A.

Fluctuations in the gender ratio at birth and the stillbirth rate are also likely outcomes of such infectious outbreaks, see Appendix A. The hospital bed occupancy rate in England also shows otherwise unexplained patterns, see Appendix A.

The hypothesis is that any number of unrecognized infections by the 3000 species of known pathogens is highly likely to precipitate at least part of such unexplained trends. As stated earlier, the evidence has been there for many years but has been ignored.

## 12. The Role of Government Policy

In theory, government policy is enacted to solve problems. However, conflicting objectives can sometimes generate unintended consequences. Hence the unintended consequences of policy when attempting to address complex problems [131] and somewhat insidious “policy-based evidence” when attempting to justify policy [132], as opposed to evidence-based policy [133]. At this point, we need to consider how a policy imperative, namely, the Private Finance Initiative (PFI), which commenced in the UK in 1992 [134], had a perverse effect on public hospital bed planning. The PFI initiative was initially an “off-balance sheet” method to build public infrastructure [135]. The government of the time was seeking to convince the public that national debt was being “prudently” managed [136]. Effectively, a private consortium borrowed the money, built the infrastructure, and then charged for the infrastructure and subsequent maintenance, which was paid out of revenue. The international rules of accounting were subsequently changed to recognize that such an “off-balance sheet” illusion was in fact public debt [137], and indeed a very expensive form of public debt [138,139,140,141].

Because PFI was such an expensive way to build hospitals, and other infrastructure, the government introduced rules stipulating that the schemes should be “affordable” [142]. The only way to achieve the “affordability” criteria was to effectively manipulate or fiddle with future bed demand considerably downwards. NHS managers and external Management Consultancies “complied” with the rules, and smaller hospitals were duly constructed. The role of external Management Consultancies was to construct a believable narrative about increasing efficiency and demand for “management”. The reality was that everyone in the NHS knew that the forecasts had been fiddled with and the hospitals were far too small [143,144,145,146].

The public did not possess the detailed technical knowledge to realize that the use of LOS as an “efficiency” measure is fraught with hidden dangers, see Appendix A, and that the decline in the real LOS within the NHS was far lower than was being predicted on paper—as above.

The point is that such manipulation was easily achieved, partly because there were no international benchmarks. Hence, claims by politicians that the hospitals were too big due to inefficiency could not be adequately dismissed.

The sad truth is that the high bed occupancy caused by too few beds relative to expressed demand directly generates chaos, harm, and inefficiency [20,81,82,83,84,85], thereby reinforcing the claim by politicians that the NHS is "inefficient." The inefficiency is, more correctly, policy imposed.

A recent analysis by the King’s Fund of the performance of the UK NHS relative to international peers has concluded that while a publicly funded NHS avoids the risk of catastrophic costs from falling ill, it has suffered from substantial capital (buildings and equipment) and manpower underinvestment, leading to poor health care outcomes [147].

## 13. Key Recommendations

From the above discussion and observations, several key recommendations can be listed:

The length of stay should be measured in real time. Occupied bed days with 2 decimal places should be sufficient (equivalent to hours and minutes).

The length of stay should be measured from the decision to admit, which may include time in the emergency department waiting for a bed to be found and perhaps also time waiting in an ambulance outside of the emergency department.

Occupied beds are the real measure of demand. Occupied bed days are the sum of all real-time LOS, including time spent waiting for a bed to be found or queuing outside and inside the emergency department.

Available beds can be split into two parts:

Staffed beds—the usual measure of available beds.

Physical beds—include staffed beds, plus beds that may be closed over the weekend, beds in day surgery units that are closed outside working hours, and beds held in reserve for infection control, deep cleaning, and surge capacity.

Hospital average bed occupancy should be measured as a real-time average and can be reported annually, monthly, weekly, or daily.

The average bed occupancy is not applicable at the whole hospital level but reflects the occupancy applicable to the different-sized bed pools, which are specific to types of patient care, i.e., critical care, pediatrics, neonates, adult medicine, adult surgical, oncology, hematology, etc., plus the need to separate males and females in the wards.

The optimum whole hospital occupancy is therefore the weighted average of the constituent patient-type and sex-specific bed pools.

Hospital bed planning cannot be conducted using annual averages, as this will guarantee that the bed numbers are too small. Both circadian, weekday, and seasonal cycles must be included, along with appropriate stand-by for infection control, deep cleaning, and surge capacity.

Hospital staffing should be flexible to allow for the natural fluctuations in demand that characterize the real-world environment, i.e., levels of local pollution, fluctuations in temperature (with extremes), and known and unknown infectious outbreaks.

Hospital planning should include a measure of unmet demand that may have accumulated in surgical waiting lists and waiting in an outpatient queue, i.e., Demand = Admissions + change in the number of people waiting.

Building a hospital that is too small is a gross waste of scarce resources since it creates inefficiency, poor patient care, staff burnout, and general chaos.

The use of the age-standardized mortality rate (ASMR) needs to be incorporated into the lines of bed equivalence in Figure 1, which will require further research.

It has been the author’s observation over the past 30 years that all the above are regularly ignored in England in a seemingly lemming-like urge to build publicly funded hospitals that are too small to operate at maximum effectiveness and efficiency. Lastly, Appendix A illustrates the central issue, namely, that hospital bed demand is the output of a complex system. All complex systems show unexpected outcomes.

## 14. Conclusions

The new model for acute beds appears to work well and has a relevant theoretical framework. It can also be used to compare staffing levels between countries. Both beds per 1000 deaths and beds per 1000 population give approximately similar outcomes only if the crude mortality rate is used as the x-axis. However, beds per 1000 population does not have an associated theoretical basis.

The method has been expanded to investigate the likely optimum numbers of beds and staff relating to different types of care. This feasible region depends greatly on the levels of support for outpatient and community care and the implementation of genuinely joined-up integrated care. The latter requires a high level of supporting political will, including relevant policies and a persistent process of implementation, reevaluation, and investment.

The high volatility in deaths, probably mediated by infectious outbreaks, has profound implications for the average bed occupancy rate for optimum efficiency. This is also influenced by hospital size, where smaller hospitals must operate at a lower average occupancy.

England has been used as an example of dysfunctional policy ‘management,’ leading to a level of acute services that is sliding into chaos and very low levels of bed provision.

## Figures and Tables

**Figure 1 ijerph-20-07171-f001:**
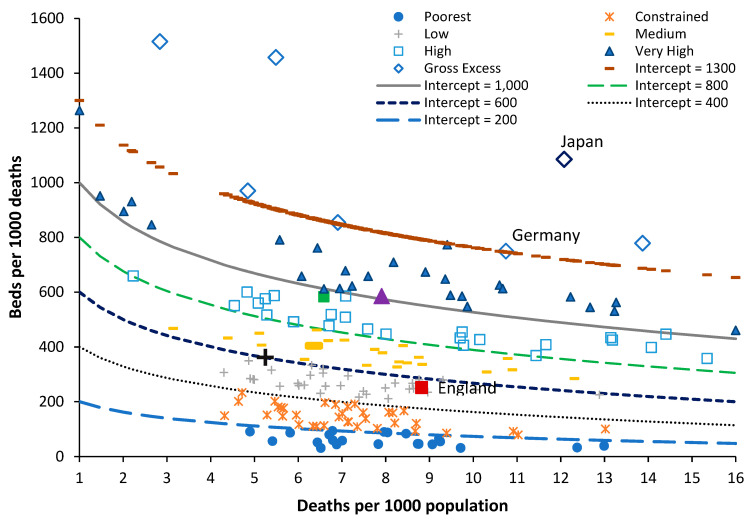
Total curative beds in world countries in 2019, including lines of equivalence. Australia, green square, England, red square (bed deficit), Switzerland, purple triangle, Germany, blue diamond, Japan, dark blue diamond, New Zealand, thick yellow dash, Malaysia, black cross. The crude mortality rate is from the World Bank [28]. The number of curative hospital beds per 10,000 population is from the World Health Organization (WHO) [29], while bed numbers and occupancy rates for European Union member states are from EuroStat [30].

**Figure 2 ijerph-20-07171-f002:**
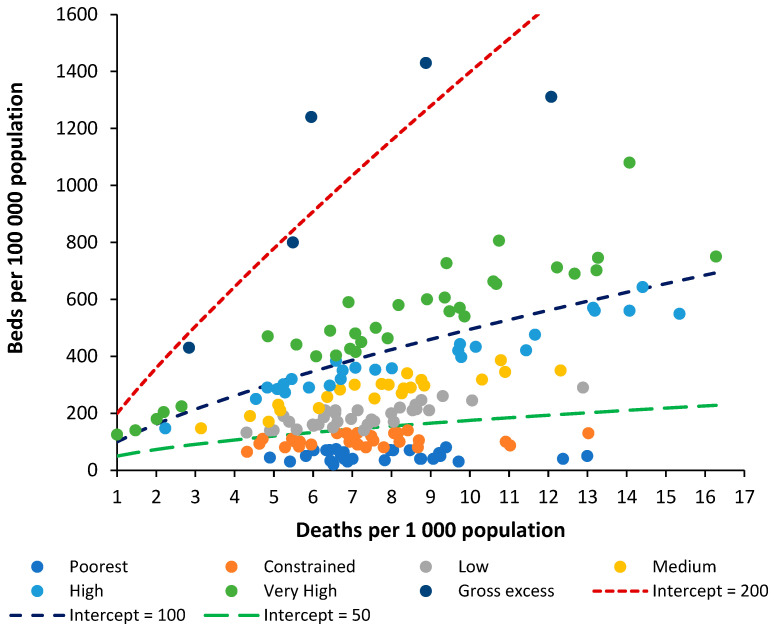
Beds per 100,000 population versus deaths per 1000 population for world countries in 2019. All data sources are as per Figure 1.

**Figure 3 ijerph-20-07171-f003:**
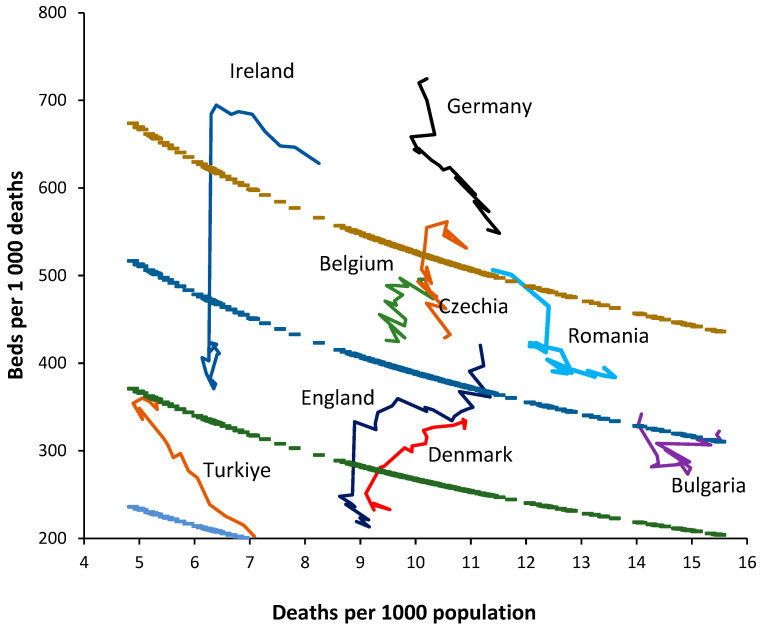
Trends in bed supply in various European countries over 20 years to 2019 and over 30 years to 2019 in England. Lines of equivalent bed supply are shown for reference. Data from [30] and that for England from [19].

**Figure 4 ijerph-20-07171-f004:**
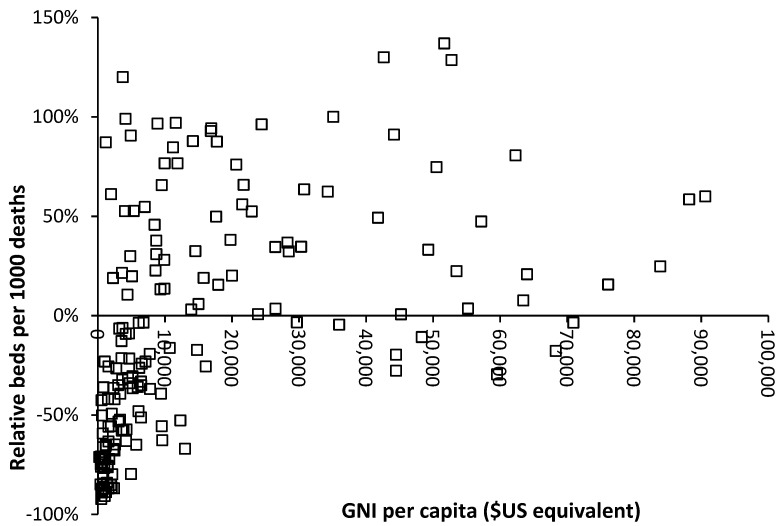
Role of gross national income (GNI) in $US equivalents on the relative number of beds. Beds are relative to the line of equivalence, with an intercept = 700. Gross national income (GNI) per capita in 2019, or extrapolated to 2019 by linear regression, is from the World Bank [33].

**Figure 5 ijerph-20-07171-f005:**
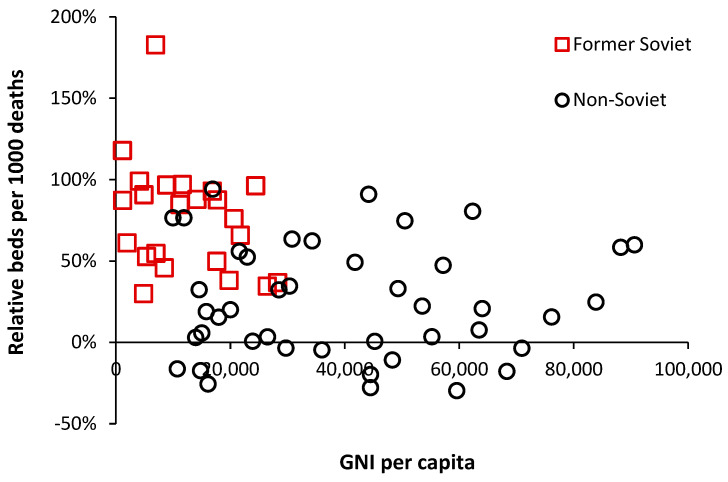
Relative bed supply in former/current Soviet bloc countries (red squares) versus their non-Soviet counterparts (circles), along with the 2019 GNI per capita. Outliers from the very high group have been excluded.

**Figure 6 ijerph-20-07171-f006:**
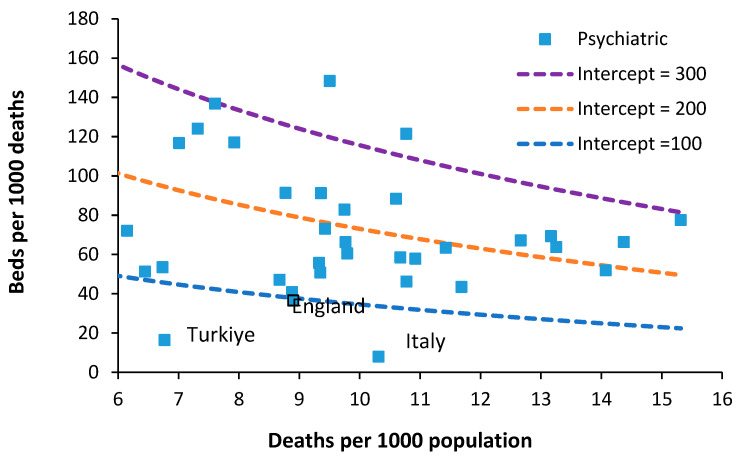
Psychiatric beds in 2019 for European countries. The data are from Europa.eu [30].

**Figure 7 ijerph-20-07171-f007:**
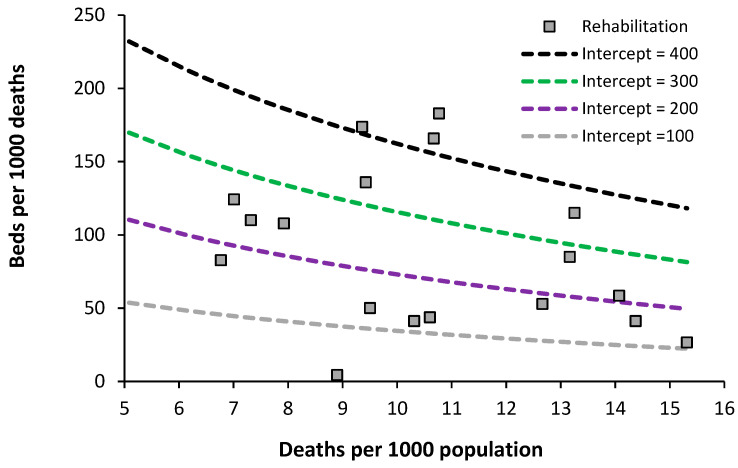
Acute rehabilitation beds in 2019 for European countries. The data is from [30].

**Figure 8 ijerph-20-07171-f008:**
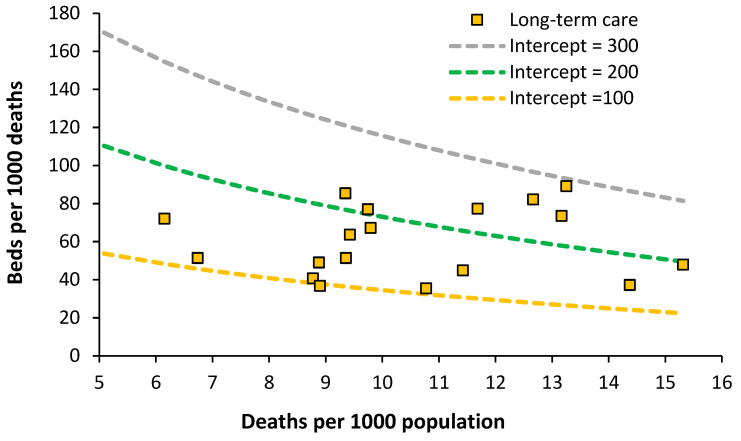
Long-term care beds in 2019 for European countries. The data is from [30].

**Figure 9 ijerph-20-07171-f009:**
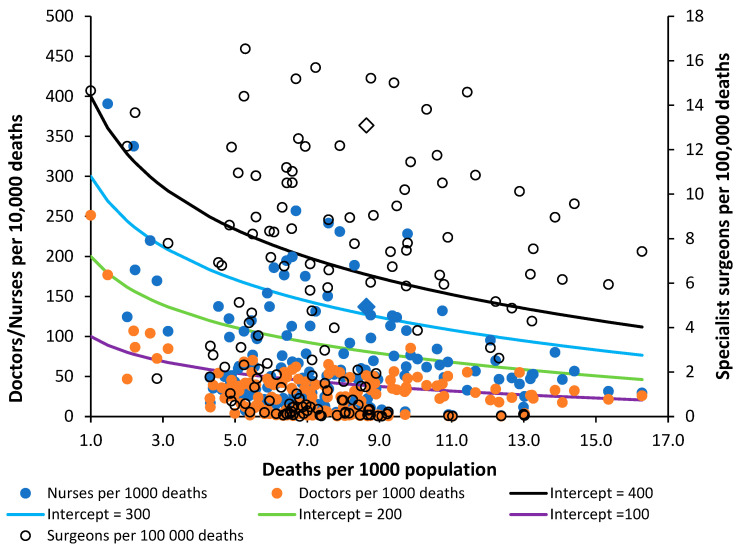
Provision of nurses, doctors, and surgeons in world countries. The data are from [38,39,40]. Blue circles are nurses per 1000 deaths, orange circles are doctors per 1000 deaths. The empty diamond is for the UK while the blue triangle is the UK.

**Figure 10 ijerph-20-07171-f010:**
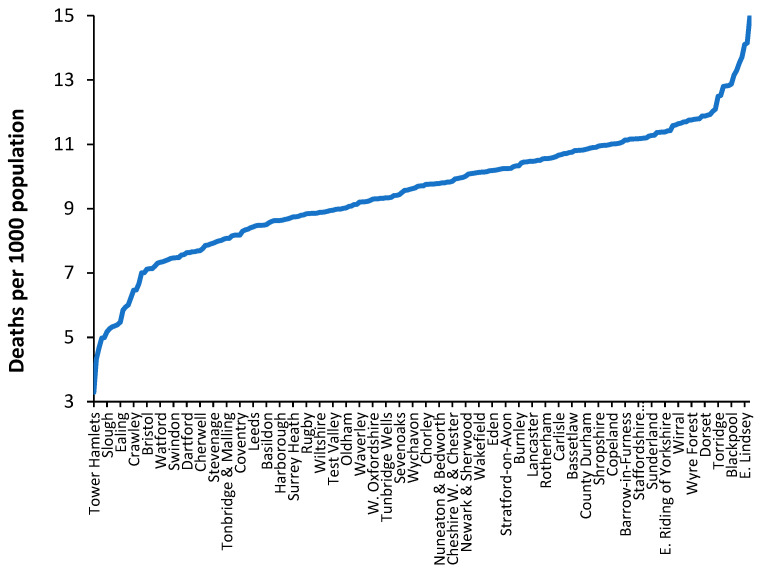
Crude mortality rate (deaths per 1000 population) for a selection of 150 English and Welsh local authorities in 2019 (the year before COVID-19). The data are from the Office for National Statistics [42].

**Figure 11 ijerph-20-07171-f011:**
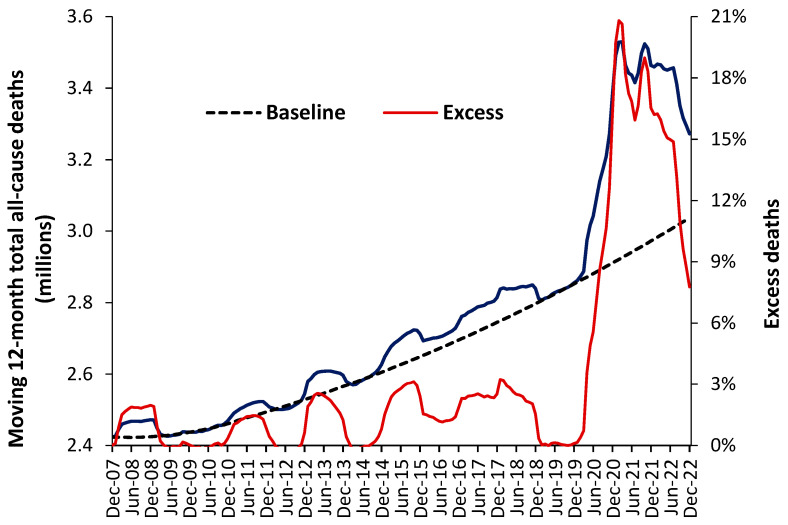
A baseline trend for total deaths in the USA using a moving 12-month total [69]. Blue line is the moving 12-month total while the dashed line is a polynomial curve fit going through the low years.

**Figure 12 ijerph-20-07171-f012:**
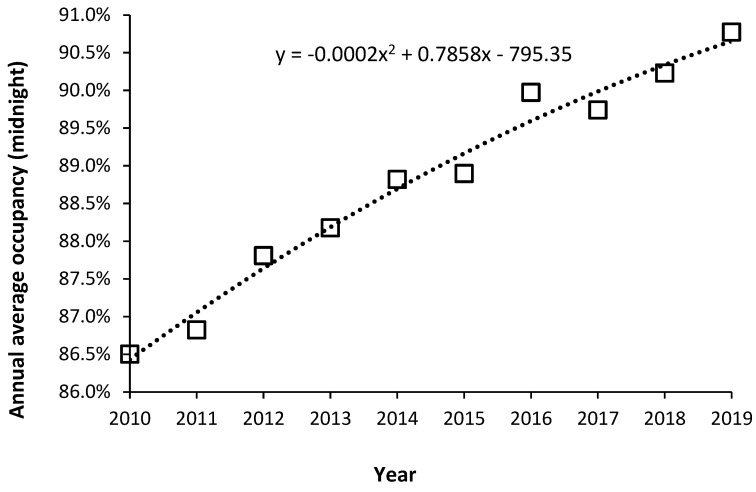
Trend in average annual midnight bed occupancy for curative beds in England from 2010 to 2019 (calendar year averages). The data are from NHS England [19]. Squares are the actual data; the dotted line is a polynomial curve fit.

**Figure 13 ijerph-20-07171-f013:**
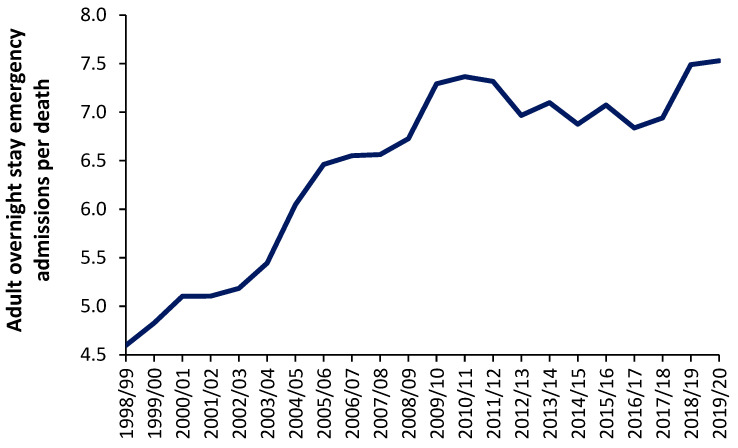
The ratio of overnight stay (adult) acute emergency admissions per all-cause deaths in England between 1998/99 and 2019/20 Data are from NHS Digital [3], while deaths are from the Office for National Statistics [95].

**Figure 14 ijerph-20-07171-f014:**
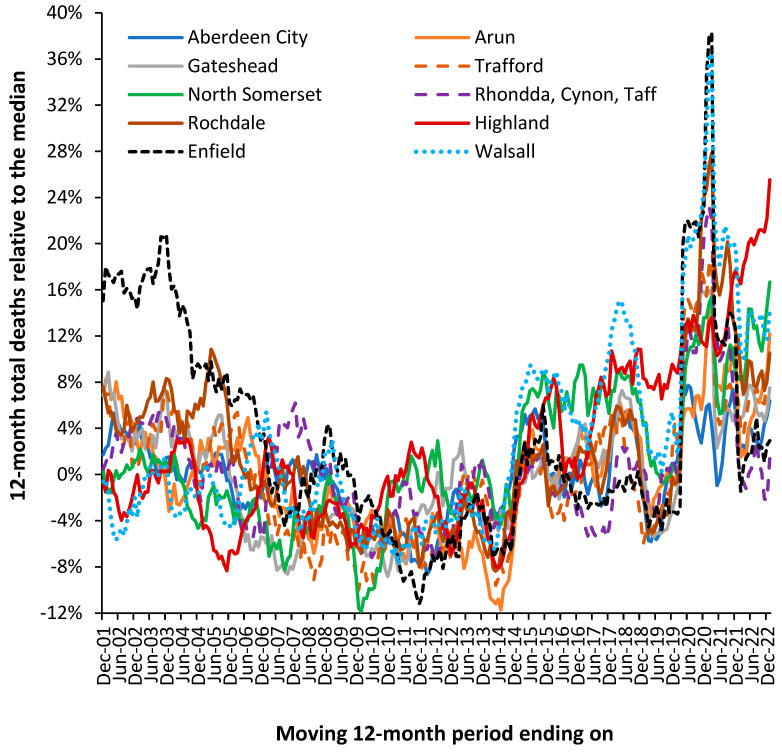
Moving 12-month total of deaths, relative to the median value, from 2001 to 2022 for a random selection of 10 local authorities in England, Scotland, and Wales. The selection criteria were between 2100 and 2600 total deaths in 2001. Monthly deaths for local authorities are from the Office for National Statistics (ONS) [95].

**Figure 15 ijerph-20-07171-f015:**
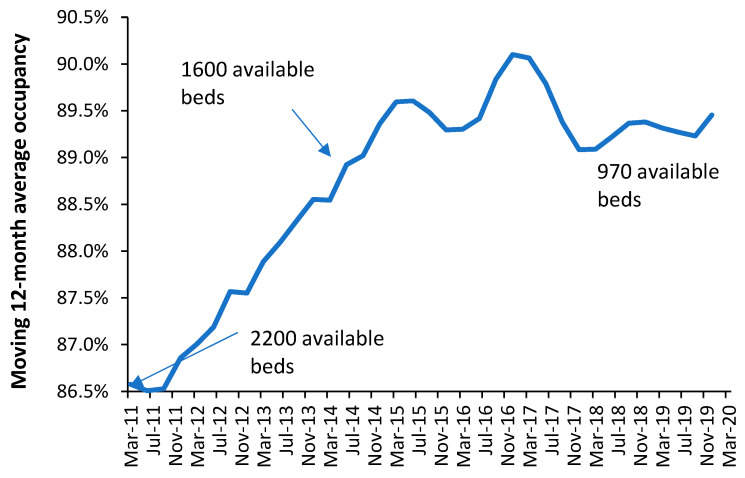
Moving 12-month average occupancy for mental illness hospital beds in England. Data from NHS England [19]. Note that the cycle commences again in late 2019 but is interrupted by the infection control measures implemented during COVID-19.

**Figure 16 ijerph-20-07171-f016:**
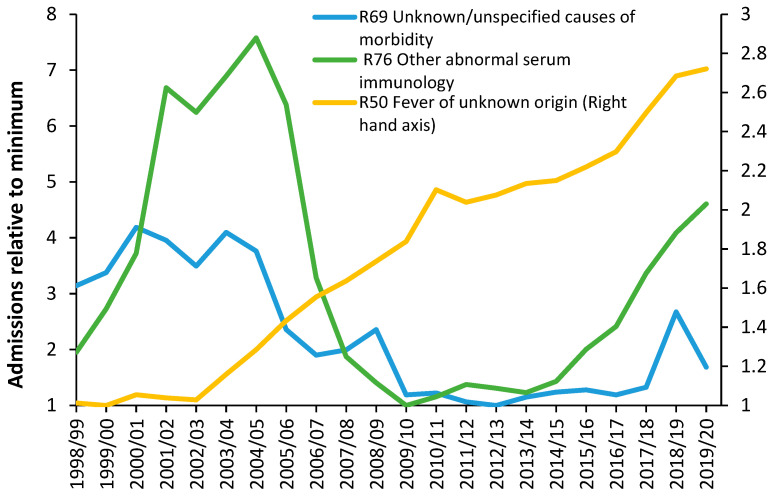
There are three examples of unexplained trends in hospital admissions in England. The data are from Hospital Episode Statistics [19].

**Table 1 ijerph-20-07171-t001:** Distribution of 17 common persistent human DNA viruses in human organs/tissues.

Count	17	15	14	13	12	11	11	11	10	6
Virus	Body	Hair	Skin	Colon	Liver	Lung	Blood	Kidney	Heart	Brain
HHV-6B	**97%**	10%	23%	**87%**	**90%**	**87%**	55%	**84%**	45%	19%
B19V	87%	13%	**87%**	74%	77%	84%	71%	**84%**	**84%**	**24%**
TTV	87%	13%	16%	42%	77%	52%	**81%**	45%	29%	13%
EBV	81%	17%	16%	39%	39%	58%	45%	52%	10%	3%
HPV	80%	**78%**	20%	9%			10%			
HHV-7	77%	23%	16%	65%	45%	61%	26%	23%	13%	
MCPyV	58%	53%	23%	10%	3%	6%		3%	6%	10%
JCPyV	48%	17%	16%	16%	23%	23%	19%	39%	13%	10%
HCMV	26%	3%	3%	6%	10%	23%	6%	6%		
HPyV6	26%	20%	13%	6%			3%			
HSV-1	19%	20%	6%	3%	3%	3%	3%	3%	3%	
BKPyV	16%	7%	3%	3%	6%		3%	3%	3%	
HPyV7	6%	7%								
HPyV10	6%	7%								
HSV-2	3%	3%								
VZV	3%		3%		3%	3%			3%	
HBV	3%		3%	3%	3%	3%		3%		

Adapted from [103]. Figures in bold are the maximum value for each tissue while the yellow highlight.

## Data Availability

All the data is publicly available.

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
