# Peer review of "Addressing the Knowledge Deficit in Hospital Bed Planning and Defining an Optimum Region for the Number of Different Types of Hospital Beds in an Effective Health Care System"

_ijerph, 2023, doi:10.3390/ijerph20247171_

Round 1

Reviewer 1 Report (Previous Reviewer 1)

Comments and Suggestions for Authors

I thank the author for considering my comments. The manuscript is, in my view, ready for publication.

Author Response

Thank you for your very valuable input and time.

Reviewer 2 Report (Previous Reviewer 2)

Comments and Suggestions for Authors

I have already revised this paper and gave my comments. I proposed the author shorten the paper because it is too long, not well organized and comprehensively described, and too philosophical and chaotic in structure. I also stressed that there are many redundant sections and subsections (The case for unrecognized pathogen outbreaks, the effect of the World War II baby boom, forecasting future deaths and population, flexible staffing regulates costs not the physical number of beds, delayed transfers of care). Instead, the author made the paper longer (from 33 to 36 pages plus supplementary material 33 pages). The goal of the paper is not clearly stated at the end of the introduction section. There is first background and then an introduction section! What is the difference between the mentioned sections? The methodology to perform the review analysis is not explained in the separate section. Instead, each section now contains a description of methods. Regarding reviews, I am not familiar with the current structure of the paper. I see that the author put a lot of effort and time into writing the paper and I would not like the manuscript to be rejected because of my opinion. So, the best solution would be to find another reviewer. For me, it is very difficult to follow and understand what is written in the current paper.

Author Response

It is always somewhat challenging when one reviewer is pleased with the paper and suggests that it be published while another struggles with elements of the presentational aspects.

The goals for the paper have been slightly lengthened as suggested.

In response to the reviewer’s initial comments the structure of the paper underwent major revision. I must disagree with the opinion that the structure is chaotic. The suggested redundant sections are in my opinion there for a purpose which is explained in each section.

It is important to note that the review introduces new concepts into the field and hence serves as a repository for further research. Hence, the somewhat extensive Supplementary material with associated description of methods. As a repository of new concepts, it is important that the paper is not shortened as it should hopefully serve as the starting point for further research in Masters and PhD theses. The concepts will hopefully be further developed and refined in the years ahead.

As stated, the idea of using Age Standardized Mortality Rate (ASMR) is simply an extension of my personal search for reliable (and readily available methods) for international comparison. No review methodology is capable of such synthesis.

I had also indicated previously that my extensive publications in this field contain probably > 1,000 unique references to other studies, i.e., the study is already backed by extensive literature review.

In terms of a structured review the work of Ravaghi et al 2020 has already admirably achieved this task, and is indeed, recommended in the study.

As to philosophical elements I removed one section as requested. I would prefer to say that the study presents a series of testable hypotheses.

As both a reviewer and academic editor I am happy with the work and suggest that developments in the field of hospital bed modelling would be hindered if it were not published.

This manuscript is a resubmission of an earlier submission. The following is a list of the peer review reports and author responses from that submission.

Round 1

Reviewer 1 Report

Comments and Suggestions for Authors

The paper is a very comprehensive, extensive manuscript concerning hospital planning in England. It presents a qualitatively improved method for determining an optimal range of hospital beds. Publication is strongly encouraged. It is an important contribution to the discussion.

The assignment to the review category of the journal may mislead readers. Namely, it may initially give the impression that it is a literature review, which is not the case. Rather, it is a compilation of various research papers and findings by the author. This should be made clear in the abstract.

I have some notes and remarks.

Major:

The paper is very extensive in terms of content and also confusing due to the lack of a formal structure of background, data and methods, results and discussion. This is noticeable, for example, in the fact that important indicators are not presented and explained in a dedicated section, but rather "somewhere". It is recommended to insert a corresponding section, possibly in section 3. It is essential to explain not only how the most important indicators are calculated, but also what they mean in the model used here and why they are used.

The figures should be better explained, and close to the figures.

The new model is applied to different sectors of health care such as psychiatric beds. What seems a bit odd is that palliative care is not depicted/discussed given the explicitly depicted nearness to death of hospitalizations. The more palliative care is developed, the lower the number of deaths in hospital and the higher the number of beds per 1000 deaths should be. In an international comparison, this is certainly an important aspect.

Minor

Row 126, 142: double explanation of the abbreviation

Row 254: "As stated earlier, the evidence has been there for many years but has been ignored." This sentence is in a way superfluous. The sentence before postulates a hypothesis. In my opinion, the sentence in question invalidates the hypothesis. Moreover, a similar statement has been made before (row 152f).

The fact that the countries of the former Eastern Bloc became independent from the SU about 20 years ago is not accurate. Rather, it is already more than 30 years. It may be helpful to note that these were centrally planned economies in which health care resources were also centrally planned.

Figure 11 makes it clear to me that it is unfortunate that there is no legend for the countries. This would be very helpful because of the different rehab systems.

Confusing for me is the term "acute rehab beds" (which implies that there are also non-acute rehab beds).

Section 11 reveals the political motivation of the paper presented and reads like a polemic in places. This seems interesting, but the question is whether this is appropriate for a scientific paper.

Author Response

Please check with the attachment.

Reviewer 2 Report

Comments and Suggestions for Authors

The review is about addressing the knowledge deficit in hospital bed planning and proposing a new approach to bed modeling which plots beds per 1000 deaths against deaths per 1000 population (The number of hospital beds per 1000 population is a commonly used indicator to compare international bed numbers). There is also attemption to define the optimum region for bed supply in an effective health care system. The goal of the paper is nowhere stated. To the best of my knowledge reviews offer a comprehensive analysis of the existing literature within a certain field of study, but here there is a lack of that literature analysis, that is, the author rather writes about his opinion and observations (related to the situation in the UK and a hospital where he was working) than cite proper scientific literature (there are many inappropriate references e.g. when he is talking about pathogens). The methodology to perform mentioned analysis is not explained at all. In my opinion, this review is not well organized, it is too philosophical and chaotic in structure. The structure includes an introduction, many redundant sections and subsections (like the case for unrecognized pathogen outbreaks, the effect of the World War II baby boom, forecasting future deaths and population, flexible staffing regulates costs not the physical number of beds, delayed transfers of care), and conclusion. There are 27 pages without supplementary material and references (in total 33)! According to IJERPH instructions a suggested minimum word count for reviews is 4000 words. I don't know what is the maximum word count but it is too long. To conclude, the paper does not have enough scientific merit to be published in its current form.

Author Response

Please check with the attachment.
